# Hydrophobic gating in bundle-crossing ion channels: a case study of TRPV4

Jian Huang [1] & Jianhan Chen [1✉]

Transmembrane ion channels frequently regulate ion permeation by forming bundle crossing of the pore-lining helices when deactivated. The resulting physical constriction is believed to serve as the de facto gate that imposes the major free energy barrier to ion permeation. Intriguingly, many ion channels also contain highly hydrophobic inner pores enclosed by bundle crossing, which can undergo spontaneous dewetting and give rise to a "vapor barrier" to block ion flow even in the absence of physical constriction. Using atomistic simulations, we show that hydrophobic gating and bundle-crossing mechanisms co-exist and complement one and another in the human TRPV4 channel. In particular, a single hydrophilic mutation in the lower pore can increase pore hydration and reduce the ion permeation free energy barrier by about half without affecting the bundle crossing. We believe that hydrophobic gating may play a key role in other bundle-crossing ion channels with hydrophobic inner pores.

[1] Department of Chemistry University of Massachusetts Amherst, Amherst, MA 01003, USA. ✉email: jianhanc@umass.edu

on channels are integral membrane proteins that provide nanoscale transmembrane (TM) pathways to facilitate ion permeation across the lipid bilayer[1]. These channels play critical roles in ion homeostasis[2,3], electric potential regulation[4], neuronal and neuromuscular transmission and cell division[5]. Central to the ion channel function is the ability to open and close the pathway in response to various stimuli to regulate ion permeation, which is referred to as gating. Studying how an ion channel gates is critical to understand its function under various physiological conditions. Such efforts have greatly benefited from a huge increase of high-resolution structures of ion channel proteins in recent years thanks to the emergence of cryo-EM technology[6,7]. Structural studies reveal that many ion channels contain a physical constriction in their "inner pore" region below the selectivity filter, which is formed by bundle crossing of the pore-lining TM helices[8,9]. This bundle-crossing gating mechanism was observed in the very first atomistic ion channel structure, that of KcsA potassium channel in the closed state[10], whereas the bundle is splayed open to release the constriction, such as in the $Ca^{2+}$-bound open state of the homologous MthK channel[11]. Voltage-gated Kv channels are also believed to use the bundle crossing mechanism to control their opening and closing under various physiological conditions[10–17]. Curiously, many ion channels lack bundle crossing and remain physically open in the closed states[18–20]. Instead, the nanoscale inner pores of these channels are sufficiently hydrophobic such that they can undergo spontaneous dewetting transition[21–23]. The resulting dry vapor region provides a large free energy barrier for ion permeation, effectively closing the pathway. These hydrophobic gating channels include BK[21,24,25], TWIK-1 channel[22], as well as many others that are activated by a wide range of stimuli, including voltage, mechanical force, and/or ligand binding[18,19].

Interestingly, many ion channels have been observed to contain a highly hydrophobic inner pore enclosed by helix bundle crossing, such as NavAb[26,27], MthK[28] and all TRP channels with known structures[29,30]. This observation raises an important question on whether bundle crossing and hydrophobic gating mechanisms can co-exist and complement one and another within the same channel. There could be synergistic effects between these two otherwise distinct gating mechanisms, that bundle crossing creates a narrower inner pore to promote hydrophobic dewetting transition and the dewetting transition further drives pore collapse to stabilize the bundle crossing and strengthen the physical constriction. In this work, we investigate this question by focusing on TRPV4, a member of TRPV subfamily that plays important roles in many biological processes, such as thermo-sensation[31,32], osmoregulation[33–37], cell swelling[38], bone homeostasis[39], nociception[40,41], and cancer cell migration[42]. TRPV4 has also been implicated in many diseases[43–45] and is considered a promising drug target[46–48]. TRPV4 is a calcium-permeable nonselective ion channel that can be activated by several stimuli of strikingly different nature, including heat[49–51], chemical ligands[52–55], mechanical force and osmotic stress[56,57]. As shown in Fig. 1a, the functional TRPV4 channel has a common tetrameric architecture[7] shared by all other TRPV members[29]. Each monomer contains an ankyrin repeat domain (ARD) that consists of six N-terminal ankyrin repeats on the cytoplasmic side. The pore domain (PD) contains pore-forming TM helices S5-6 and harbors the selectivity filter, and the voltage-sensing-domain-like (VSDL) domain consists of TM S1-4. Note that PDs and VSDLs are domain swapped (Fig. 1a). The selectivity filter or "the upper gate" of TRPV4 has a large opening of ~7 – 8 Å in diameter (Fig. 1b), allowing hydrated metal ions to pass freely[58]. The pore-lining S6 helices from four subunits bundle together to create a narrow constriction near M718 (human TRPV4, or hTRPV4, numbering), forming "the lower gate". The inner pore region between the lower and upper gates is highly hydrophobic and lined exclusively with hydrophobic residues (Fig. 1b). The co-existence of physical constriction and nanoscale hydrophobic inner pore suggests that both bundle crossing and hydrophobic gating could contribute to TRPV4 deactivation. An agonist-bound open conformation of hTPRV4 is also available, which shows that the S6 helices undergoes α-to-π transition at V708[59]. The resulting S6 rotation exposes a set of more hydrophilic residues towards the pore center, makes I715 to form the narrowest constriction instead of M718 and enlarges the pore size of the bundle-crossing lower pore region[59].

In this work, we performed atomistic molecular dynamics (MD) simulations in explicit solvent to investigate the hydration property of TRPV4 pore in the closed state, and further determined the free energy profile of $K^+$ permeation to identify the location of the actual gate. To dissect the contributions of hydrophobic gating and bundle crossing to channel gating, we analyzed the pore hydration and $K^+$ permeation-free energy properties of a hydrophilic and non-pore-facing mutation I715N near the bundle crossing, which was experimentally found to increase the resting channel activity of TRPV4[60]. Our results strongly support that both bundle-crossing and hydrophobic gating indeed contribute to TRPV4 inactivation. Given the prevalence of hydrophobic inner pores in ion channels, the current study suggests that hydrophobic gating likely plays a more general role than previously thought, regardless of the presence of bundle crossing in the deactivated channel.

## Results

**Deactivated hTRPV4 readily undergoes dewetting transition in the lower pore region.** In this work, we focus on hTRPV4 due to its biomedical significance. A homology structure model of hTRPV4 in the deactivated state was derived from that of the highly homologous xTRPV47, which has a sequence identity of 86% in the TM region (see Methods). The model reveals that the inner pore of hTRPV4 is lined exclusively with hydrophobic residues (Fig. 1b). The HOLE pore profile analysis shows that the narrowest constriction of the closed state locates in M718 and has a diameter of ~2 Å, insufficient for a hydrated $K^+$ to cross. The inner pore region overall resembles an inverted cone with a maximum diameter of ~12 Å near I703, in principle providing a large physical cavity for water molecules and ions. Yet, previous simulations showed that hydrophobic cylindrical nanopores can readily undergo dewetting transition when the pore diameter is reduced below ~10 Å[18,61]. Such dewetting transitions have also been observed in biological pores such as those of BK and TWIK-1 K2P channels with diameters of ~10 Å[21,22]. Therefore, it is likely that the inner pore of hTRPV4 can undergo the hydrophobic dewetting transition to create a vapor barrier inaccessible to bulk water and ions.

To examine the hydration properties of the hTRPV4 pore, we performed multiple independent 500 ns atomistic simulations in explicit solvent and membrane, with and without positional restraints on $C_\alpha$ atoms (see Methods). All simulations were initiated with a state where the whole pore region was fully hydrated with water molecules (Fig. 2a). Note that the lower pore, defined as the region between the planes of Cα atoms of V708 and M718, has enough physical volume to accommodate approximately 40 water molecules (Fig. 2b insert). In our unrestrained simulations, the lower pore region almost immediately underwent a dewetting transition and formed a dry vapor region in all three repeated simulations of wild-type (WT) hTRPV4 (Fig. 2a, Supplementary Fig. 1 and 2A). Note that restraining the pore conformation to the starting Cryo-EM structure does not affect the dewetting transitions (Supplementary Fig. 1, 2 and 8). In all simulations, the dewetting transitions occurred within a few ns, and the number of waters in the lower pore region remained below ~10 for the rest of the simulations. In contrast, the upper pore

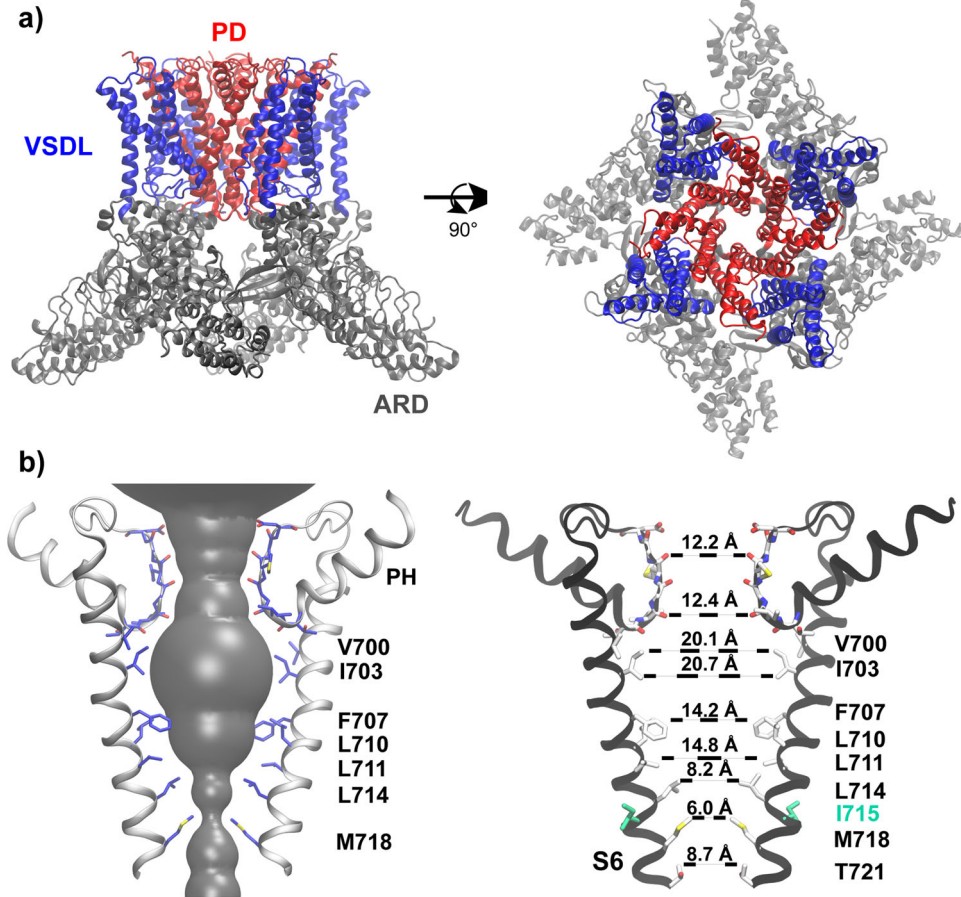

**Fig. 1 Structural features of human TPRV4 (hTRPV4). a** Homology model of hTRPV4 in the closed state shown in the front and top views. The structure was built based on the cryo-EM structure of *Xenopus Tropicalis* TRPV4 (PDB: 6BBJ)[7] (see Methods for details). The Ankyrin repeat domain (ARD), voltage-sensing-domain-like domain (VSDL) and pore domain (PD) are colored in gray, blue and red, respectively. **b** Ion permeation pathway of hTRPV4, with only the S6, pore helix (PH) and selectivity filter loop from two opposing subunits shown for clarity. Key pore-lining residues are shown in sticks and labeled. The pore opening profile (the grey surface on the left panel) was calculated using HOLE[85]. The nearest heavy atom distances from opposing residues are measured on the right panel. Carbon, oxygen, nitrogen, and sulfur atoms are colored in white, red, blue and yellow, respectively.

region (between V700 and V708), with the presence of polar groups from the selectivity filter, remained hydrated throughout the simulations, even though the hydration water number decreased slightly from ~100 and stabilized at ~70 (Fig. 2b). The average water density along the membrane normal quantifies the level of dehydration of hTRPV4 (Fig. 3b, blue), which clearly indicates a nearly 14 Å-long vapor region ranging from about −4 Å to 10 Å. The extended vapor region in the lower pore should provide an additional free energy barrier to ion permeation, a characteristic of the hydrophobic gating mechanism.

**Both bundle-crossing and vapor region contribute to ion permeation barrier**. To further analyze the contributions of the vapor region and bundle-crossing to hTRPV4 gating, we performed umbrella sampling simulations to calculate the free energy profile of $K^+$ permeation through the pore. Analysis of the ion distribution from equilibrium simulations (see Methods) show that the single ion region spans around −20 to 20 Å, beyond which there are non-negligible probabilities of observing two or more ions (Supplementary Fig. 3). The latter is particularly true near the extracellular entrance (~40 Å), which is lined by a ring of negative-charged D682 and E684. As such, we perform umbrella sampling only in the single ion region (Supplementary Fig. 4) and combine the resulting PMF with the one derived directly from equilibrium simulations to construct the final PMF (see Methods), as illustrated in

Supplementary Figure 5. The $K^+$ permeation PMF for the WT hTRPV4 shows a significant free energy barrier of ~18 kcal mol⁻¹ (Fig. 3c, blue trace), consistent with a deactivated channel state. Importantly, the major barrier located near the bundle crossing site, which is consistent with a major role of physical constriction in channel deactivation. Curiously, the barrier does not peak at the narrowest point of the pathway near ~ −2 Å (lower green dash line in Fig. 3a). Instead, the maximum free energy barrier shifts upwards towards the vapor barrier, to ~ 0 Å, and the overall barrier is broad, spanning ~ −3 to 6 Å. These features indicate that the dewetted lower pore region supplements the bundle crossing to enforce the deactivated state of hTRPV4. We further analyze the water coordination number of K+ during permeation. As shown in Fig. 4, the coordination number is only reduced to ~4.5 from the bulk value of ~ 6.5 throughout the lower gate region. That is, the ion remains highly hydrated during permeation, consistent with the notion that the vapor region provides an additional barrier to the permeating ions. Note that there exists no apparent barrier in the selectivity filter (z = ~30 Å), due to the large opening of ~8 Å in the selectivity filter region (Fig. 3a). Lack of free energy barrier in the selectivity region is consistent with the non-selective nature of the channel.

**I715N mutation reduces ion permeation barrier by increasing lower pore hydration**. It was previously discovered that hydrophilic glutamine mutation at site I715, but not others in the S6

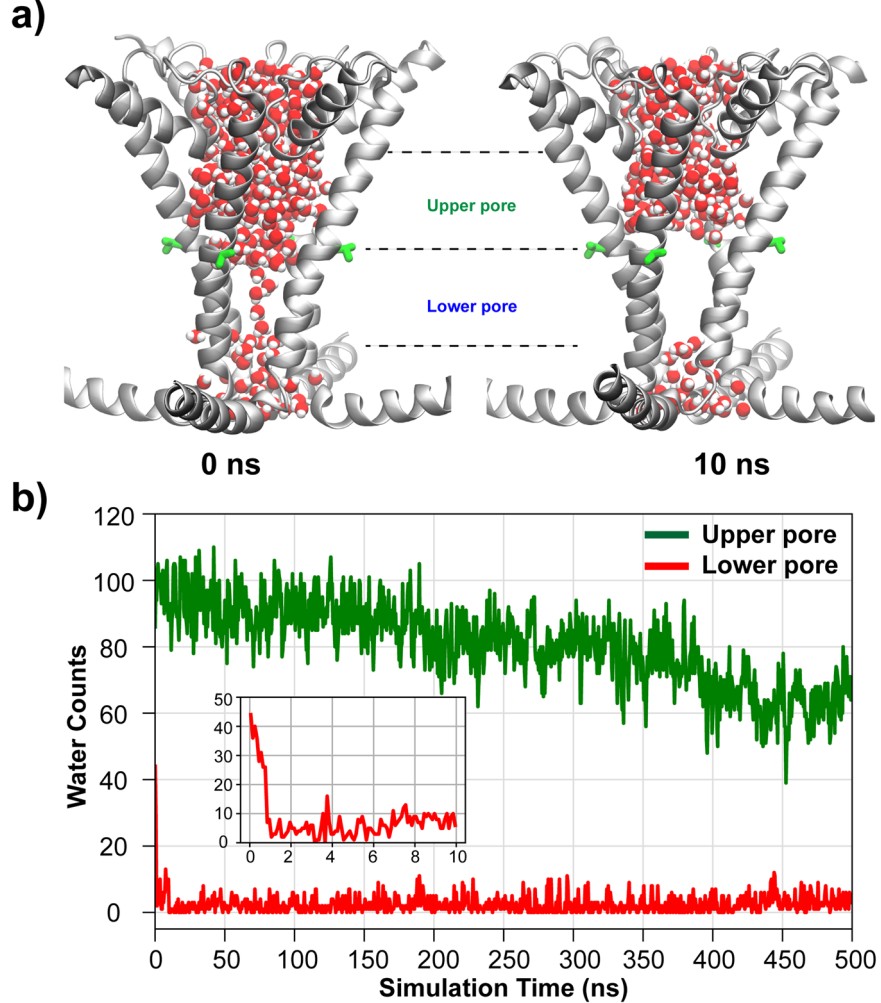

**Fig. 2 Hydration of hTRPV4 pore during atomistic simulations. a** Representative snapshots of hTRPV4 pore in wet (0 ns) and dry (10 ns) states. Only the pore helix, S6 and TRP helices are shown for clarity. Pore waters are shown using balls with oxygens and hydrogens colored in red and white, respectively. **b** Numbers of water molecules in the upper and lower pore regions as a function of simulation time during one of three unrestrained simulations. The upper and lower pore regions are defined using the Cα positions of V700, V708 and M718. The insert shows the water count in the lower pore during the first 10 ns.

helix, resulted in a dramatic increase of the basal channel activity in whole-cell recordings[60]. The I715N mutant channel could be further activated using agonist 4-α-PDD to reach the same level of channel activity as the WT[60]. The implications are that the fully activated states of WT and I715N hTRPV4 channels are similar and that I715N mutation mainly perturb the properties of the deactivated state. Interestingly, I715 is not pore-facing, and the I715N mutation does not directly disrupt the physical constriction formed by bundle crossing (Fig. 1b). We performed three independent simulations to examine how I715N may indirectly perturb the pore conformation. As summarized in Supplementary Figure 6, the mutation introduces minimal impacts on the structure and the pore remains very stable. The Cα RMSD of pore-lining S5 and S6 helices remain ~2 Å throughout the 500 ns simulations, comparable to the WT simulations. Average pore profiles calculated from the last 100 ns simulations (1 frame per ns) are very similar for the WT and I715N mutant channels (Fig. 3a). In particular, the bundle crossing region (z ~ 0 Å) remains essentially identical with the same narrowest constrictions. Interestingly, the lower inner pore region (z ~ 5 to 12 Å) appears slightly expanded, while the filter region appears slightly narrower in the I715N channel.

During the equilibrium simulations, the I715N hTRPV4 channel also readily undergoes dewetting transitions to form a vapor phase

in the lower pore region (Supplementary Figure 7). However, the vapor region appears destabilized, spanning a slightly smaller length, and the pore becomes more hydrated around the narrowest constriction point (Fig. 3b). Analysis of the probability distribution of water numbers in the lower pore region confirms an increased hydration level in the I715N channel, by ~4 waters (Supplementary Figure 8A). Similar observations can be made in restrained simulations, where all C-α atoms of the channel were harmonically restrained with respect to the reference cryo-EM structure. Restrained simulations eliminate structural relaxation in response to the mutation and allow one to focus on the mutational impact on the pore hydration due to the physical properties of Asn sidechain. The results also show an increase in pore hydration, albeit only by 2-3 waters (Supplementary Fig. 7 and 8B).

Increased hydration in the bundle crossing region could help reduce the free energy barrier of ion permeation and lead to the increased resting state channel activity in the I715N mutant hTRPV4 channel observed in whole-cell measurements[60]. To test this hypothesis, we calculate the K+ permeation PMF following the same protocol as used for the WT channel (see Methods). The result reveals a striking decrease of the free energy barrier to less than 10 kcal mol−1 in the I715N mutant channel, which reflects a nearly 50% reduction from that of the WT channel (Fig. 3c, red

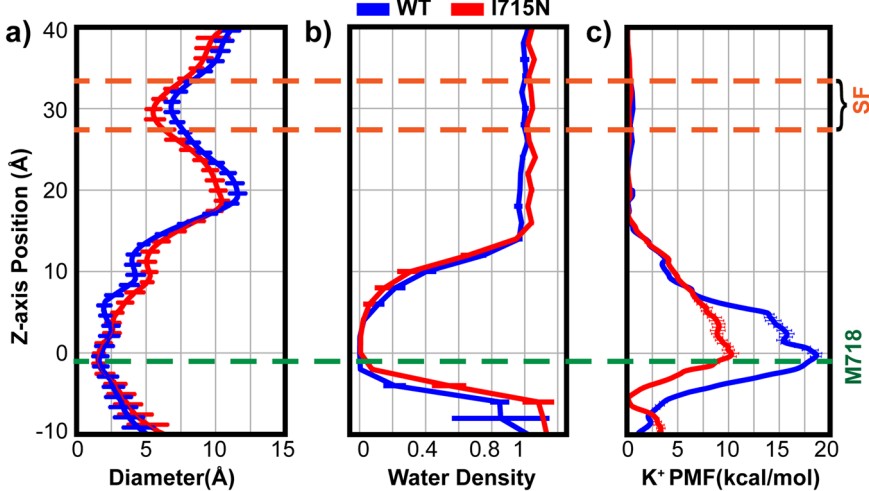

**Fig. 3 hTRPV4 pore profile, hydration propensity and K$^+$ permeation free energy in the WT and I715N systems. a** The pore diameter and **b** water density profiles were calculated using the last 100 ns trajectories of simulations of the WT and I715 mutant hTRPV4 channels (see Methods). The error bars plot the standard deviations within the trajectories. **c** The K$^+$ ion permeation PMFs were calculated from umbrella sampling simulations (for single ion region, −20 Å to 20 Å) and equilibrium MD simulations (elsewhere) (see Methods). The error bars plot the difference between results from the first and second halves of these trajectories. In all panels, the zero position is the lower gate location, defined as the backbone center of mass of residues 715−718. The yellow and green dash lines mark the z-axis positions of the selectivity filter (SF) and the narrowest constriction (near M718).

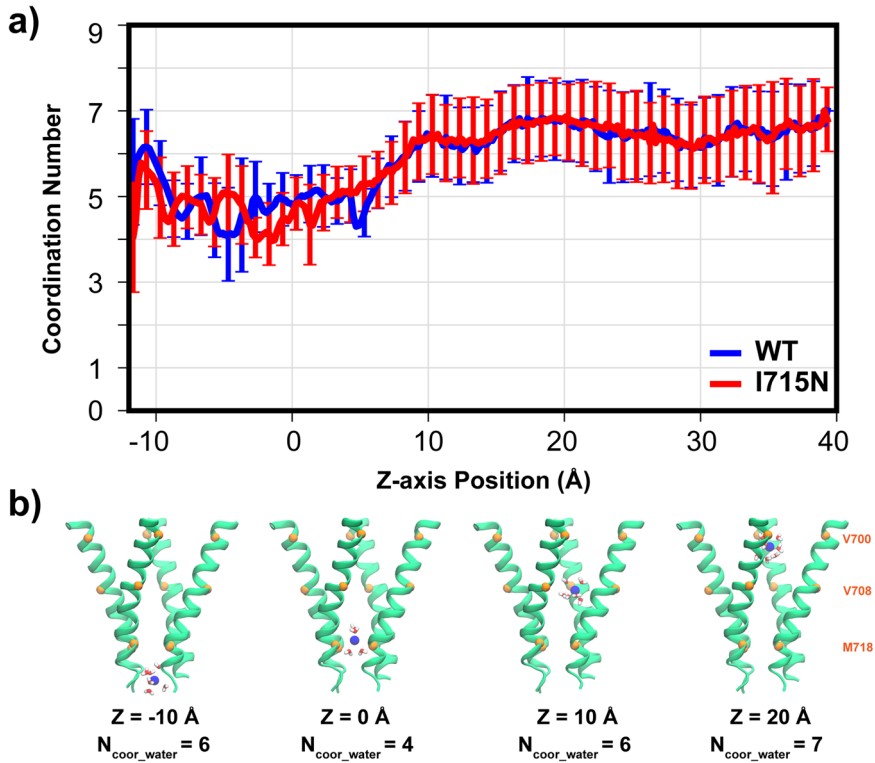

**Fig. 4 Hydration of permeating K$^+$ through hTRPV4. a** The average water coordination number is shown a function of the z-axis position, extracted from umbrella sampling simulations (see Methods). Error bars plot the standard deviations during the last 18 ns of umbrella sampling simulations, which reflect the natural fluctuation of water coordination. **b** Representative snapshots of K+ with different numbers of coordination water molecules along z-axis positions are also shown. Only S6 helices are shown as cartoons for clarity and the Cα atoms of V700, V708 and M718 marked using orange spheres.

vs. blue traces). Such a large reduction of ion permeation-free energy barrier is particularly notable because I715N has little impact on the bundle crossing constriction (Fig. 3a). That is, I715N mainly disrupts the hydrophobic vapor barrier by increasing pore hydration at bundle crossing. This notion is further tested by examining two-dimensional (2D) PMFs of ion permeation as a function of the z distance of the ion to the lower

gate and the number of waters in the lower pore region, calculated from the same umbrella sampling trajectories using WHAM. As shown by Fig. 5, the lower pore region is required to be solvated by ~20 waters at the critical barrier of ion permeation ~0 Å. In the I715N channel, the overall hydration level in the lower pore region is elevated to ~10 waters on average, compared to that of ~0 water in the WT channel. Ion needs to pay a large

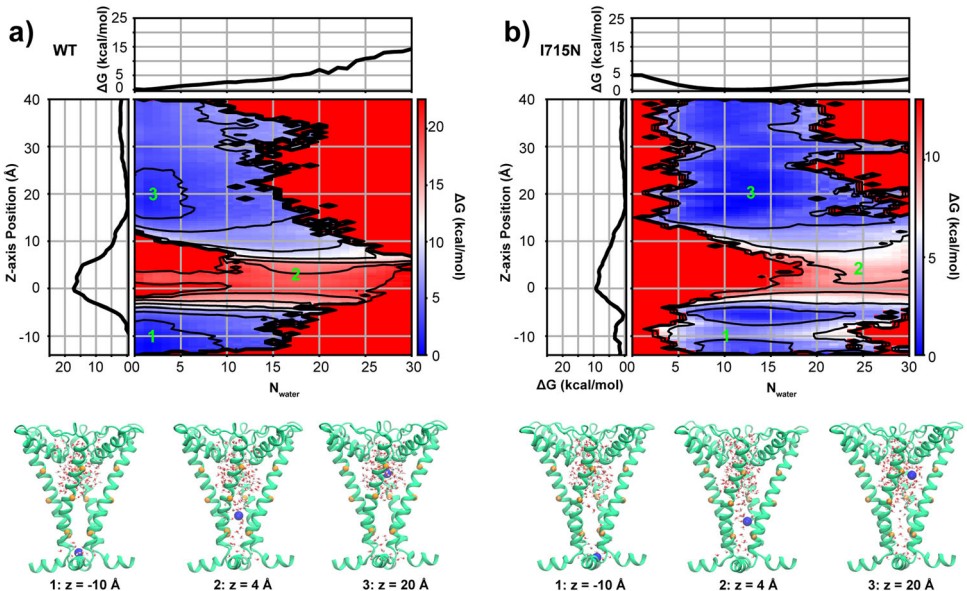

**Fig. 5 2D PMFs of the K$^+$ permeation in WT. a** and I715N mutant **b**. hTRPV4 channels, as a function of the water number in the lower pore region and z distance of the ion to the lower gate. The results were calculated from the umbrella sampling trajectories using WHAM. Both 2D PMFs were shifted just so that the minima at z ~ 20 Å is at zero. Contour lines are plotted every 3 kcal mol$^{-1}$. Representative snapshots at three critical points (−10 Å, 4 Å and 20 Å, also marked as 1, 2, 3 on the top panel) are shown on the lower panel, where only PH and S6 helices are depicted as cartoons for clarity with K$^+$ and water molecules inside the pore. The Cα atoms of V700, V708 and M718 are represented as orange spheres.

free energy cost to pass through the lower pore region where the hydration free energy of the lower pore is ~6 kcal mol$^{-1}$ cal/mol at N$_{water}$ = 20 in the WT system (Fig. 5a). However, in the I715N system hydration free energy of the lower pore has a minimum at ~12 water molecules and it only needs to cross a hydration barrier of ~3 kcal mol$^{-1}$ at N$_{water}$ = 20 (Fig. 5b). Therefore, increased hydrophilicity of the lower pore environment due to I715N mutation shifts the pore hydration-dehydration transition equilibrium to stabilize a more hydrated state and thus facilitate ion permeation. Importantly, the ~50% decrease in the overall barrier of the I715N mutant channel suggests that bundle-crossing and hydrophobic gating mechanisms contribute similarly to WT hTRPV4 deactivation. We note that the deactivated I715N channel still gives rise to a significant free energy barrier for ion permeation, which is consistent with the experimental observation that the I715N mutant could be further activated by adding the TRPV4 agonist 4-αPDD[60].

## Discussion

TM ion channels frequently contain bundle-crossing of pore-lining helices in the deactivated state, which are generally believed to be physical gate of the ion permeation pathway. Curiously, many ion channels also contain a highly hydrophobic inner pore between the bundle-crossing and selectivity filter, which will undergo a hydrophobic dewetting transition to enter a dehydrated state. The resulting dry vapor region will also impose a large free energy barrier for ion permeation. The latter is known as the hydrophobic gating mechanism and has been suggested for many ion channels that lack an apparent physical constriction in the closed state (e.g., no bundle-crossing). In this work, we examine if both bundle-crossing and hydrophobic gating mechanisms could both be at play and compliment one and another for ion channels that have been observed to contain both bundle-crossing and a highly hydrophobic pore in the closed state. Using hTRPV4 as an important model system, our atomistic simulations confirm that its lower pore region can spontaneously undergo dewetting transition and form a vapor phase. Free energy analysis reveals that the major free energy barrier for ion permeation locates near the bundle crossing site but

is broad and spans the entire dry region in the lower pore. This suggests that both bundle-crossing and vapor barrier contribute to the channel deactivation. We further examined the hydration and free energy profiles of mutant hTRPV4 channel, where a nonpore-facing residue I715 is replaced with the hydrophilic N mutation. The I715N mutant channel has been experimentally shown to display increased channel activity in the resting state[60]. Atomistic simulations show that I715N mutation increases pore hydration without affecting the bundle-crossing constriction of the pore. Strikingly, the mutation leads to nearly ~50% reduction of ion permeation-free energy barrier, despite essentially identical levels of physical constriction in the WT and mutant channels. The implication is that bundle-crossing and hydrophobic gating contribute about equally in gating of hTRPV channels. We note that the free energy barrier in I715N still appears to be too large to explain the observed resting activity[60]. The implication is that I715N may lead to additional conformational relaxation near the constriction site that are not captured in the current 500 ns simulations. It should also be noted that the hydration properties of small cavities can be sensitive to the specific explicit solvent force field of choice, especially when the traditional nonpolarizable ones[20,62–64].

Many ion channel structures have now been observed to contain both bundle-crossing and a hydrophobic pore[26–30] and can be expected to benefit from both gating mechanisms as illustrated for hTRPV4 in this work. The conformational transitions involved in bundle-crossing formation is often accompanied with changes in the pore environment. For example, in TRPV family proteins[30,65], the S6 helix can either adopt a stable π-helix functioning as a "hinge" in the upper region to allow helix motions to open the pore (e.g., TRPV1[66] and TRPV5[67,68]) or switch between α-helix and π-helix during opening process (e.g., TRPV3[69] and TRPV6[70]). Those movements involve the S6 rotation, exposing a different set of side chains to the pore and modulating the pore hydrophobicity and geometry. The synergistic interplay between bundle crossing and hydrophobic gating may enable more versatile regulation of channel activities. Mutations, ligand binding or post-translational modifications could shift water dynamics in the pore region and control the ion permeation-free energy barrier without significantly

changing the pore structure. Therefore, hydrophobic gating likely plays an even more prevalent role in ion channel gating than currently thought.

## Methods

**Homology modeling of hTRPV4 in the closed state**. The closed state structure of hTRPV4 was derived from the cryo-EM structure of the closed state of xTRPV4 (PDB: 6BBJ)[7], which share 78% sequence identity for the whole protein and 86% for the TM region. Multisequence alignment between xTRPV4 (uniprot accession: A0A6I8RQZ3) and hTRPV4 (uniprot accession: Q9HBA0) was performed through Tcoffee web server (https://tcoffee.crg.eu/) (Supplementary Note 1). The alignment result was then used in MODELLER[71] to build the homology model of deactivated hTRPV4. The final model contains residues 148-639 and 661-786. The long loop of residues 640-660 and the N and C-terminal tails were not resolved in the template structure and presumed dynamic. These dynamic regions were thus not in the current simulations. The terminal residues at all truncation sites were capped with either acetyl groups (for N-termini) or N-methyl groups (for C-termini). The initial closed conformation of hTRPV4 I715N mutant was constructed directly from the WT homology model by replacing the original sidechain with a suitable rotamer of the new residue that minimizes steric clashes.

**Atomistic simulations**. Models of the closed-state conformation were first embedded in independently generated 1-palmitoyl-2-oleoyl-sn-glycero-3-phosphatidylcholine (POPC) bilayers, and then solvated with TIP3P water using the CHARMM-GUI web server[72–74]. 150 mM KCl was added to buffer and neutralize the systems. The final simulation boxes consist of ~57,000 TIP3P water molecules, ~450 lipids and ~220 ions, with a total ~270,000 atoms and a dimension of ~$13.6 \times 13.6 \times 14.3$ nm$^3$. The protein was described using the CHARMM36m protein force field[75] and lipids using the CHARMM36 lipid force field[76]. All simulations were performed using GPU-accelerated GROMACS 2019[77,78]. The nonbonded forces were calculated with a cutoff of 12 Å and a smoothing switching function starting at 10 Å. Electrostatic interactions were calculated using the Particle Mesh Ewald (PME) algorithm[79]. The SHAKE algorithm was used to constrain all hydrogen-containing bonds[80]. The system temperature was maintained at 298 K using the Nose-Hoover thermostat[81,82]. The system pressure was maintained semi-isotropically at 1 bar at the x-y plane using the Parrinello-Rahman barostat algorithm[83].

We performed 3 parallel simulations under unrestrained conditions, where no other extra restraint was added, and 3 parallel simulations under restrained conditions, where all C$_\alpha$ atoms of the protein were restrained using a force constant of 2.39 kcal mol$^{-1}$ Å$^{-2}$ (or equivalently, 1000 kJ/(mol·nm$^2$)) with respect to the reference configuration of the hTRPV4 homology model mentioned above. The restrained simulations aim to evaluate if hydrophobic dewetting transitions depend on the ability of the pore to contract during simulations. Both unrestrained and restrained systems undergo the same equilibration and production procedures. For equilibration, all systems were first subjected to 5000 steps of energy minimization using the steepest descent algorithm and then underwent a series of equilibration steps where the heavy atoms of protein and lipids (and ligands) were harmonically restrained with the restraining force constant gradually decreasing from 10 to 0.1 kcal mol$^{-1}$ Å$^{-2}$, as recommended by CHARMM-GUI[74]. All normal production runs were performed under the NPT condition (constant particle number, constant pressure and constant temperature) with 2 femtoseconds as the MD integration time. To minimize the effect of the missing loop (640-660), C$_\alpha$ atoms of the two ends residues (639 and 661) were harmonically restrained to their initial positions using a force

constant of 1.0 kcal mol$^{-1}$ Å$^{-2}$ in all simulations. The I715N system was directly created from the WT system through CHARMM-GUI by mutating the 715 site. Then, a similar equilibration, three runs of restrained as well as unrestrained simulations were performed.

**Umbrella sampling protocol for ion permeation-free energy profile**. The potential of mean force (PMF) profiles of K$^+$ permeation through the pore were calculated using umbrella sampling[84] for both WT and I715N hTRPV4. The initial structures of these systems were taken from representative fully equilibrated snapshots from the equilibration simulations described above. The initial conformations for each umbrella sampling window were generated using steered MD simulations, during which C$\alpha$ atoms of the protein were harmonically restrained using a force constant of 2.39 kcal mol$^{-1}$ Å$^{-2}$ (equals to 1000 kj mol$^{-1}$ nm$^{-2}$)). During the steered MD simulations, a pulling harmonic potential was applied to a pre-selected potassium ion within the pore region with a force constant of 1.91 kcal mol$^{-1}$ Å$^{-2}$ (equals to 800 kj mol$^{-1}$ nm$^{-2}$)). The initial conformations were then carefully inspected to ensure minimal protein structural distortions. Umbrella sampling windows covered the whole TM region, ranging from −20 Å to 20 Å with respect to the bundle crossing site (COM of the backbone of residue 715 to 718), with an interval of 2 Å (see Supplementary Figure 4). Harmonic restraints with a force constant of 2.0 kcal mol$^{-1}$ Å$^{-2}$ were applied to the selected potassium ion along the z-axis during umbrella sampling and each window was simulated for 20 ns. Sampling from the last 18 ns included the weighted histogram analysis method (WHAM) analysis to calculate the both 1D and 2D PMF profiles. Note calculation of 2D PMF along K$^+$ z-position and pore hydration water count does not require biasing in the second dimension, because pore hydration can readily reach equilibrium in each umbrella window along the K$^+$ z-position. Uncertainty of PMF profiles was estimated by calculating standard errors between the first half and the second half of the last 18-ns umbrella sampling. Analysis of potassium occupancy along the Z-axis during the equilibrium simulations (see above) revealed that there is a high probability of double potassium occurrence near the selectivity filter (~30 Å above the bundle-crossing gate) (see Supplementary Figure 3). To avoid multiple ion regions, we decide that the single-ion PMF profiles from umbrella sampling were thus only valid from −20 Å to 20 Å. PMFs in regions outside of this range were derived from normal 500 ns equilibration simulations: firstly, K$^+$ ion counts along z-axis with a cutoff radius of 10 Å and a bin size of 2 Å (on the z-axis) were calculated; secondly, those counts were converted to probabilities to get the PMFs. We stitched the whole potassium permeation PMF profiles by combining umbrella sampling (range: −20 Å – 20 Å) and normal 500 ns equilibration simulations (range: −80 Å – −20 Å and 20 Å – 54 Å) through linear interpolation (see Supplementary Fig. 5).

**Analysis**. To calculate average pore profile, we extracted the snapshots every 1 ns during the last 100 ns of one of the three parallel 500-ns trajectories. All snapshots were first aligned using all C$\alpha$ atoms and the pore profiles then calculated using HOLE[85] and averaged to produce the final pore profiles shown. When calculating water density along the Z-axis (Fig. 3b), a cutoff of 2 Å was used to count water number occurrence within a bin cylinder by slicing Z-axis. The final density was further normalized such that the bulk solvent region has a density of 1.0. The pore region of hTRPV4 is divided into the upper pore and lower pore regions using the plane defined by C$\alpha$ atoms of V708 residues (Fig. 2b). The regions for counting water numbers are approximated using

truncated cones, giving that it resembles maximally the pore shape (Fig. 1). Cα atoms from residues V700 and V708 of four monomers define the upper circle and lower circle of the upper pore truncated cone, whereas Cα atoms of residues V708 and M718 define the circles of the lower pore truncated cone. A cutoff distance of 3.5 Å between ion and water oxygen was used for calculating the water coordination number of a selected K+ [86].

**Statistics and reproducibility**. All standard MD simulations were performed in triplicates to quantify the standard errors in all plots. The convergence of the umbrella sampling free energy simulations was assessed using block analysis.

**Reporting summary**. Further information on research design is available in the Nature Portfolio Reporting Summary linked to this article.

## Data availability

All data supporting the findings of this study are available within the paper and its Supplementary Information. The source data file for all main figures could be found with the following link: https://doi.org/10.6084/m9.figshare.24309802.v1. Any remaining information can be obtained from the corresponding author upon reasonable request.

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

## Acknowledgements
We thank Erik Nordquist and Zhiguang Jia for insightful discussions. This work is supported by the National Institutes of Health through grant R35 GM144045 (to JC). This research was partially supported by a fellowship from the University of Massachusetts as part of the Chemistry-Biology Interface Training Program (National Research Service Award T32 GM139789) (to JH).

## Author contributions
J.H. and J.C. conception and design of the study; J.H. performing simulations and analysis; J.H. and J.C. analysis and interpretation of the data, drafting and revising the manuscript.

## Competing interests
The authors declare no competing interest.
