## [Peer Review File · Communications Biology]

Reviewers' comments:

Reviewer #1 (Remarks to the Author):

This is a careful atomistic MD and US simulation of the TRPV4 channel focusing on the role of the hydrophobic gating in the ion permeation barrier. They conducted multiple simulations for two molecular models: WT and I715N mutant since the ion permeation recording results between them are very different. They obtained lower PMF for K⁺ permeation in the I715N mutant and resolved the mechanisms. They showed the importance of the increase of pore hydration in the I715N mutant compared to WT.

The simulations and analysis were done carefully and the paper is well written for readers to understand what they found easily. I have almost no concerns about the manuscript for publication.

One thing that I am interested in is Figure S3, which shows one, two, and three potassium ions along the z-axis in the central pore region of WT hTRPV4. How the behaviour is changed in the I715N mutant? This question is related to the role of double and triple ions in the pore for estimating PMF in such complex structures.

Also, I would like to know how technically the double or triple ions are avoided in PMF calculations.

But, these two are rather minor issues, I think.

Reviewer #2 (Remarks to the Author):

The manuscript "Hydrophobic gating in bundle-crossing ion channels: a case study of TRPV4" by Jian Huang and Jianhan Chen reports the MD study of hydrophobic gate properties of closed TRPV4 channel for two cases: WT and I715N mutant. The hydrophobic gates and the vapor phase occurring there is a hot topic of modern research of ion channels gating. The manuscript contains high-quality simulation results and is of interest. In particular: numerical evaluation of the effect of hydrophilic substitution in the hydrophobic gate of TRPV on the energy barrier for the ion passing, that can be helpful for further investigations of hydrophobic gating phenomena. However, in my opinion, the obtained energy barrier of about 10 kcal/mol in I715N mutant is still too high to explain the experimentally observed resting activity of this mutant (ref [60]). The authors should add some further suggestions about possible mechanisms of weakening of the hydrophobic gate in the light of their research. I also have some specific comments, as outlined below.

Major comments:

1 (Page 9) "However, in the I715N system hydration free energy of the lower pore has a minimum at ~ 12 water molecules and it only needs to cross a hydration barrier of ~3 kcal/mol at $N_{\text{water}}=20$ (Figure 5B upper panel)": I see on the 2d map of the panel that the minimal energy path from Z=30Å to Z=-10Å goes through the barrier of about 10 kcal/mol. Why does 3 kcal/mol of the top curve make sense?

2 The authors make an hTRPV4 model from xTRPV4 by homology modelling. But there isn't an explanation why. The experimental xTRPV4 structure would be a more sustainable model for the goals of the study. The same question applies to K⁺ ions instead of "default" for MD-simulations Na⁺.

Minor comments:

1 (Page 5) "... independent 500 ns atomistic explicit solvent simulations in explicit solvent and membrane ...": doubling of "explicit solvent".

2 Capture of Fig. 2: the descriptions of panels A and B are mixed up.

3 (Page 7) "Analysis of the ion distribution from equilibrium simulations (see Method)": Method -> Methods

4 (Page 12) "To minimize the effect of the missing loop (640-660), Ca atoms of the two ends residues

(639 and 661) were harmonically restrained using a force constant of $1.0 \text{ kcal}/(\text{mol} \cdot \text{\AA}^2)$ in all simulations.”: Were positions of the atoms or distances between the atoms restrained?
5 (Page 12) “Harmonic restraints with a force constant of $2.0 \text{ kcal}/(\text{mol} \cdot \text{\AA}^2)$ were applied to the selected potassium ion during umbrella sampling”: Were restraints applied along the z-axis or along all three axes?

Reviewer #3 (Remarks to the Author):

In their manuscript, Huang and Chen address an important question in the field of ion channel gating, that is the role of hydrophobic gating. The authors use molecular dynamics simulations in a model system (TRPV4) to get microscopic view of the role of both physical barrier provided by the bundle-crossing, as well the energetic barrier provided by the dehydration process. By studying an experimentally proposed I715N, which destabilizes the deactivated/closed state, the authors showed that the barrier for ion permeation can be reduced by half, without the bundle-crossing destabilization. The study is overall well-designed and executed, and the conclusions are supported by the data. I feel however that there are several points that should be addressed before I can recommend the manuscript for publication.

Major comments:

It is not clear how 2D PMFs were calculated. The authors mention they used the same simulations as used for the 1D PMFs, where only the z-position of an ion was restrained. How was then the second CV biased (or was it biased at all?) that is the number of water molecules in the pore? Please explain. In the unrestrained simulations the number of water molecules in the pore region seems to be below 10 (Figure 1), whereas 2D PMFs seem to sample numbers above 15. How was such a number obtained? Is it due to the ion presence which is fixed in the hydrophobic region and attracts water molecules? If so, some conformational changes around helix-bundle crossing should be expected? Please explain. Finally, the plots are somewhat difficult to read. Adding a few snapshots illustrating the critical points of 2D PMFs, showing both ion position and water molecules in the pore, would help the reader navigating the free energy surfaces.

It is known that the behaviour of water molecules in small protein cavities in MD simulations depends critically on the force field (see Lynch et al., ACS Nano 2021 and other papers from Mark Sansom lab). In the current work, I tend to believe that the authors capture the correct trend, by using the experimentally-verified mutation, and that the MD results are in accord with experiments. Ideally however, I'd like to see whether a completely different force field (say one of the AMBER flavours or OPLS) capture the same trend as CHARMM36 used here. I realize that this might be beyond the resources/time constraints, so I won't push for these additional simulations - the authors should however discuss the role of the force field and the potential caveats, especially when using fixed-charge models.

The authors mention that the studied mutation, I715N, does not lead to major changes in the structure of the helix-bundle crossing, but one simulation shows a relatively large increase in the RMSD (Fig S6 A, black trace), please explain.

Minor comments

Figure 2, captions for A and B swapped

Figure 3 B, please provide error bars

Figure 4, I think it would help combining this figure with some snapshots showing potassium ion hydration in a few selected regions along the permeation pathway

(Original reviewer comments are shown in blue)

Reviewer #1 (Remarks to the Author):

This is a careful atomistic MD and US simulation of the TRPV4 channel focusing on the role of the hydrophobic gating in the ion permeation barrier. They conducted multiple simulations for two molecular models: WT and I715N mutant since the ion permeation recording results between them are very different. They obtained lower PMF for K⁺ permeation in the I715N mutant and resolved the mechanisms. They showed the importance of the increase of pore hydration in the I715N mutant compared to WT.

The simulations and analysis were done carefully and the paper is well written for readers to understand what they found easily. I have almost no concerns about the manuscript for publication.

Responses: We greatly appreciate the reviewer for the enthusiastic support.

One thing that I am interested in is Figure S3, which shows one, two, and three potassium ions along the z-axis in the central pore region of WT hTRPV4. How the behaviour is changed in the I715N mutant?

Responses: This is a great question. The inner pore region of the I715N mutant remains similarly dehydrated (Fig 3B), and we expected that the distribution of ions along the Z-axis would be very similar to that of the WT channel. Here, we calculated the actual ion distributions for the I715N mutant channel and validated that they are indeed similar to those of the WT channel (see **revised Fig S3**).

(Revised Figure S3)

This question is related to the role of double and triple ions in the pore for estimating PMF in such complex structures.

I would like to know how technically the double or triple ions are avoided in PMF calculations.

But, these two are rather minor issues, I think.

Responses: These are great questions. Indeed, we need to avoid double or triple ions in umbrella sampling PMF calculations. This was exactly why we must first analyze the ion distributions and only apply umbrella sampling in the “single-ion” region(s). The final PMF was constructed by combining results from umbrella sampling and equilibrium simulations.

The overall procedure is explained on Page 7, “*Analysis of the ion distribution from equilibrium simulations (see Methods) show that the single ion region spans around -20 to 20 Å, beyond which there are non-negligible probabilities of observing two or more ions (Figure S3). The latter is particularly true near the extracellular entrance (~40 Å), which is lined by a ring of negative-charged D682 and E684. As such, we perform umbrella sampling only in the single ion region (Figure S4) and combine the resulting PMF with the one derived directly from equilibrium simulations to construct the final PMF (see Methods), as illustrated in Figure S5.*”

Reviewer 2

The manuscript “Hydrophobic gating in bundle-crossing ion channels: a case study of TRPV4” by Jian Huang and Jianhan Chen reports the MD study of hydrophobic gate properties of closed TRPV4 channel for two cases: WT and I715N mutant. The hydrophobic gates and the vapor phase occurring there is a hot topic of modern research of ion channels gating. The manuscript contains high-quality simulation results and is of interest. In particular: numerical evaluation of the effect of hydrophilic substitution in the hydrophobic gate of TRPV on the energy barrier for the ion passing, that can be helpful for further investigations of hydrophobic gating phenomena. However, in my opinion, the obtained energy barrier of about 10 kcal/mol in I715N mutant is still too high to explain the experimentally observed resting activity of this mutant (ref [60]). The authors should add some further suggestions about possible mechanisms of weakening of the hydrophobic gate in the light of their research. I also have some specific comments, as outlined below.

Responses: We thank the reviewer for the positive comments and support. We largely agree with the reviewer that the 10 kcal/mol free energy barrier in the I715N mutant may still be too large to explain the experimentally observed resting activity. We think that the I715N mutation may trigger additional conformational relaxation at the constriction site that was not captured during our simulations. Along this line, we have added the following note at the end of the first paragraph of our discussion section (**Page 11**):

“... We note that the free energy barrier in I715N still appears to be too large to explain the observed resting activity (60). The implication is that I715N may lead to additional conformational relaxation near the constriction site that are not captured in the current 500 ns simulations.”

Major comments:

1 (Page 9) “However, in the I715N system hydration free energy of the lower pore has a minimum at ~ 12 water molecules and it only needs to cross a hydration barrier of ~ 3 kcal/mol at $N_{\text{water}}=20$ (Figure 5B upper panel)”: I see on the 2d map of the panel that the minimal energy path from $Z=30\text{\AA}$ to $Z=-10\text{\AA}$ goes through the barrier of about 10 kcal/mol. Why does 3 kcal/mol of the top curve make sense?

Responses: The reviewer is correct that the 2D map of Fig 5B shows a minimal free energy path for **ion permeation** with a barrier of ~ 10 kcal/mol. The statement in question is actually referring to **hydration free energy**, which is the projection of the 2D free energy map on the N_{water} coordinate. The lower free energy barrier of hydration (compared to the apparent higher ion permeation free energy barrier) is a direct consequence of integration of the free energy surface along the ion position coordinate (which is often referred to as “hidden barriers”).

2 The authors make an hTRPV4 model from xTRPV4 by homology modeling. But there isn't an explanation why. The experimental xTRPV4 structure would be a more sustainable model for the goals of the study. The same question applies to K^+ ions instead of “default” for MD-simulations Na^+ .

Responses: This is a good question. Our overall research project focuses on understanding human TRPV4 (the construct studied in our experimental collaborator's lab), even though the current study could be performed on xTRPV4. Nonetheless, it should be noted these two channels are highly homologous and that the role of hydrophobic gating in bundle-crossing ion channels should be similar. We add a brief note to the manuscript stating “**In this work, we focus on hTRPV4 due to its biomedical significance.**” (page 5, beginning of Results).

Regarding the choice of Na⁺ vs K⁺ for bulk ions, K⁺ is more convenient for examining the ion distributions (e.g., **Figure 3S**) and it is required for constructing the overall PMFs by combining bulk K⁺ distributions and results from umbrella sampling (e.g., see **Figure S5**).

Minor comments:

1 (Page 5) “... independent 500 ns atomistic explicit solvent simulations in explicit solvent and membrane ...”: doubling of “explicit solvent”.

Responses: Thank you for pointing this out. We have removed the first “explicit solvent” from the main text.

2 Capture of Fig. 2: the descriptions of panels A and B are mixed up.

Responses: The descriptions of panel A and B have been corrected.

3 (Page 7) “Analysis of the ion distribution from equilibrium simulations (see Method)”: Method -> Methods

Responses: The typo has been corrected.

4 (Page 12) “To minimize the effect of the missing loop (640-660), C α atoms of the two ends residues (639 and 661) were harmonically restrained using a force constant of 1.0 kcal/(mol \cdot \AA^2) in all simulations.”: Were positions of the atoms or distances between the atoms restrained?

Responses: We have updated the text to clarify that these are positional restraints: “... C α atoms of the two ends residues (639 and 661) were harmonically restrained **to their initial positions** ...”

5 (Page 12) “Harmonic restraints with a force constant of 2.0 kcal/(mol \cdot \AA^2) were applied to the selected potassium ion during umbrella sampling”: Were restraints applied along the z-axis or along all three axes?

Responses: Yes, this is correct. We have revised the text to clarify that: “Harmonic restraints with a force constant of 2.0 kcal/(mol \cdot \AA^2) were applied to the selected potassium ion **along the z-axis** during umbrella sampling and each window was simulated for 20 ns.”

Reviewer 3

In their manuscript, Huang and Chen address an important question in the field of ion channel gating, that is the role of hydrophobic gating. The authors use molecular dynamics simulations in a model system (TRPV4) to get microscopic view of the role of both physical barrier provided by the bundle-crossing, as well the energetic barrier provided by the dehydration process. By studying an experimentally proposed I715N, which destabilizes the deactivated/closed state, the authors showed that the barrier for ion permeation can be reduced by half, without the bundle-crossing destabilization.

The study is overall well-designed and executed, and the conclusions are supported by the data. I feel however that are several points that should be addressed before I can recommend the manuscript for publication.

Responses: We thank the reviewer for the strong support!

Major comments:

It is not clear how 2D PMFs were calculated. The authors mention they used the same simulations as used for the 1D PMFs, where only the z-position of an ion was restrained. How was then the second CV biased (or was it biased at all?) that is the number of water molecules in the pore? Please explain.

Responses: The second CV (number of pore water molecules) is not biased during umbrella sampling. WHAM analysis can be used to process resulting 2D histograms as long as the correct biasing potentials are provided ($k = 2.0$ kcal/mol/Å² in first dimension and 0.0 in the second dimension). Convergence of the resulting free energy surface will depend on sufficient sampling of unbiased, spontaneous fluctuation of pore water within each umbrella window along the first dimension (z-position of K⁺).

We have added the following statement in Methods to clarify (**Page 13**): “. **Note calculation of 2D PMF along K⁺ z-position and pore hydration water count does not require biasing in the second dimension, because pore hydration can readily reach equilibrium in each umbrella window along the K⁺ z-position.**”

In the unrestrained simulations the number of water molecules in the pore region seems to be below 10 (Figure 1), whereas 2D PMFs seem to sample numbers above 15. How was such a number obtained? Is it due to the ion presence which is fixed in the hydrophobic region and attracts water molecules? If so, some conformational changes around helix-bundle crossing should be expected? Please explain. Finally, the plots are somewhat difficult to read. Adding a few snapshots illustrating the critical points of 2D PMFs, showing both ion position and water molecules in the pore, would help the reader navigating the free energy surfaces.

Responses: Yes, it is due to the ion presence during umbrella sampling. Restraining ion near the bundle crossing and in the lower pore region attracts water molecules and (artificially) increase the hydration level. We have updated **Figure 5** in include three critical points/snapshots as suggested (also see below)

(revised Figure 5)

It is known that the behaviour of water molecules in small protein cavities in MD simulations depends critically on the force field (see Lynch et al., ACS Nano 2021 and other papers from Mark Sansom lab). In the current work, I tend to believe that the authors capture the correct trend, by using the experimentally-verified mutation, and that the MD results are in accord with experiments. Ideally however, I'd like to see whether a completely different force field (say one of the AMBER flavours or OPLS) capture the same trend as CHARMM36 used here. I realize that this might be beyond the resources/time constraints, so I won't push for these additional simulations - the authors should however discuss the role of the force field and the potential caveats, especially when using fixed-charge models.

Responses: We fully agree with the reviewer that the force field itself could have a large impact on the hydration property of small cavities. As suggested, we have added a discussion on the role of force field and the potential caveats and cite a few relevant references (**page 11**): “**It should also be noted that hydration properties of small cavities can be sensitive to the specific explicit solvent force field of choice, especially when the traditional nonpolarizable ones (20, 84-86).**”

The authors mention that the studied mutation, I715N, does not lead to major changes in the structure of the helix-bundle crossing, but one simulation shows a relatively large increase in the RMSD (Fig S6 A, black trace), please explain.

Responses: Indeed, sim1 has a larger pore RMSD (~ 2.6 Å) compared to < 2 Å in sim2 and sim3. We checked the trajectory and compared the last frame of this simulation with the WT S5-6 (see below). The result shows us that even though it seems to be $0.5\sim 1$ Å higher than the other two simulations, the structures still align fine with the WT homology model. The larger RMSD comes from a slightly more distortion on one of the S5 helix and hydrophobic clasp of the S6 helix

(see the arrows in the top view in the left panel of the below figure), which also happens in the other I715N simulations though to a smaller extent (see the right panel). We do not consider this as a major conformational change since this slightly larger distortion/clasp only happened once out of three parallel simulations and also all parallel simulations showed similar conformational changes though to slightly different extent.

Figure: Comparison of S5-6 from the last frame of I715N simulation **sim1** (Red in left panel) and **sim2** (Green in right panel) with the WT homology model (blue in both panels)

Minor comments

Figure 2, captions for A and B swapped

Responses: it has been updated. Thanks for the careful proofreading.

Figure 3 B, please provide error bars

Responses: Figure 3B has been updated in the manuscript to include error bars.

(revised Figure 3)

Figure 4, I think it would be help combining this figure with some snapshots showing potassium ion hydration in a few selected regions along the permeation pathway

Responses: Four representative snapshots at three points (-10 Å, 0 Å, 10 Å and 20 Å) have been selected for the WT system and included in the lower panel of **revised Figure 4** (also see below).

(revised Figure 4)

REVIEWERS' COMMENTS:

Reviewer #2 (Remarks to the Author):

I am satisfied with the revision of the manuscript. The authors clarified all my points and the manuscript is suitable for publication in my opinion. Only a few minor concerns can be added:

- 1 (Page 7, Line 175) "(lower green dash line in Figure 3A)": there isn't any green line on the figure.
- 2 There is not any mention of how "PMFs derived from distributions from unrestrained simulations" were calculated.

Reviewer #3 (Remarks to the Author):

All my comments were addressed sufficiently

(Original reviewer comments are shown in blue)

Reviewer #2 (Remarks to the Author):

I am satisfied with the revision of the manuscript. The authors clarified all my points and the manuscript is suitable for publication in my opinion. Only a few minor concerns can be added:

Responses: We greatly appreciate the reviewer for the enthusiastic support.

1 (Page 7, Line 175) “(lower green dash line in Figure 3A)”: there isn’t any green line on the figure.

Responses: Thank you so much for pointing out. The green line has been added to the Figure 3A in the main manuscript (also see below).

(The revised Figure 3)

2 There is not any mention of how “PMFs derived from distributions from unrestrained simulations” were calculated.

Responses: The information of “PMFs derived from distributions from unrestrained simulations” was indeed missing in the previous manuscript. We have added the details in the Methods part:

“To avoid multiple ion regions, we decide that the single-ion PMF profiles from umbrella sampling were thus only valid from -20 Å to 20 Å. PMFs in regions outside of this range were derived from normal 500 ns equilibration simulations: firstly, K⁺ ion counts along z-axis with a cutoff radius of 10 Å and a bin size of 2 Å (on the z-axis) were calculated; secondly, those counts were converted to probabilities to get the PMFs.”